# Sample-Efficient Behavior Cloning using Hierarchical Event Memory of Bayesian Networks

## Abstract

When humans imitate others they often rely on their memory of individuals demonstrating the desired behaviors to emulate. This not only permits people to reproduce taught behavior, but also enables them to generalize procedural expertise to novel situations. This paper introduces HEMS-BC; a behavior cloning agent that uses a novel, psychologically plausible model of event memory. From sequential observations of an expert, the memory system represents states, observations, and state transitions as Bayesian causal models, and stores them into a hierarchically organized event memory taxonomy. In response to observation queries, posterior conditional samples of observation-action pairs are drawn to rectify class imbalances in the training dataset that the imitation learner uses to approximate the expert policy. Our findings show that our method achieves and maintains expert-level performance from fewer expert demonstrations compared to a baseline system with no event memory capabilities.

## 1 Introduction

The advent of deep learning has fostered the proliferation of neural network-based learning agents capable of demonstrating, and sometimes surpassing, human-level performance on various tasks [2, 1, 11, 15]. In particular, behavior cloning (BC) has been a common choice for building intelligent agents because of its architectural simplicity, scalability, and ability to rapidly approximate policies from expert demonstrations [4, 6].

One of the key challenges, however, in employing BC is the difficulty to overcome distributional biases [3]. One recent BC approach, Positive Unlabeled Behavior Cloning (PUBC) [14], learns a classifier that extracts the ideal samples from a large dataset to refine the BC learning process. In [13], the authors use a diffusion model to model expert state-action pairs, which is then used to guide policy learning by denoising and generating data for BC. In our approach, *HEMS-BC*, we leverage the Hybrid Event Memory System (HEMS) [8, 9, 7] to represent states, observations, and environment dynamics as Bayesian networks, and store them in a hierarchically organized event memory taxonomy. HEMS-BC uniformly draws posterior conditional samples of observation-action pairs to create a balanced training dataset that the BC learner uses to approximate the expert policy.

This work makes two novel contributions. First, it incorporates the use of Bayesian networks to generate balanced datasets for behavior cloning. Second, it introduces a novel representation for state transition models that abstract observation and state content from the environmental dynamics.

Submitted to Workshop on Bayesian Decision-making and Uncertainty, 38th Conference on Neural Information Processing Systems (BDU at NeurIPS 2024). Do not distribute.

## 2 The Hybrid Event Memory System

We previously introduced the Hybrid Theory of Event Memory[8] which commits to six theoretical postulates on the nature of event memory: 1) Event memory is a long-term memory that stores episodes and schemas; 2) Episodes are propositional representations of specific events; 3) Schemas are first-order propositional templates with probabilistic annotations; 4) Event memory elements are organized in hierarchies; 5) Retrieval cues play a central role in remembering; and 6) Remembering an event involves performing structural matching and probabilistic inference.

HEMS implements this theory by using Bayesian networks [10] to represent episodes and schemas. A Bayesian network is a directed acyclic graph describing the joint probability distribution of a set of correlated variables. Formally, for variables $\mathbf{X}$; a function returning the paraents of variable $X_i \in \mathbf{X}$, $pa(X_i)$; and a set of discrete-valued assignments to the variables, $\{x_i \cup \mathbf{u}\}$, we state:

$$P(X_1, .., X_N) = \prod_{i=1}^{N} P(X_i = x_i | pa(X_i) = \mathbf{u}) \tag{1}$$

Figure 1 shows an example event hierarchy built from a number generation domain. `Episode 1` describes a situation where the value "10" is generated by the variable $Number$. Consequently, variable $Ten$ contains $P(Ten = \text{"True"} | Number = \text{"10"}) = 1$. A similar joint probability distribution exists in `Episode 2`. `Schema 1`, however, is a probabilistic generalization of both episodes, summarizing elements of both.

HEMS constructs such a hierarchy in an online manner by inserting new episodes in a top-down manner. This insertion procedure greedily sorts the new memory element through the hierarchy, choosing the branch with the highest likelihood of generating the new example. HEMS penalizes the likelihood score, considering the complexity of the model proportional to the number of episodes it summarizes, to combat overfitting. Next, HEMS merges the episode contents into the best-matching schema to update the model. After the merge takes place, HEMS recurses down the branch of the merged schema to continue insertion, setting the new episode's parent to the modified schema. The insertion process ends when the current episode parent is the best match, or when HEMS merges the new episode with another existing episode.

Additionally, HEMS includes procedures for probabilistic inference and conditional sampling. Conditional sampling utilizes the posterior network given by probabilistic inference, and probabilistic inference begins with cue-based retrieval, which returns a memory element in response to a retrieval cue. A retrieval cue specifies a (partially) observed episode that HEMS uses to identify the best-matching element. In similar fashion to insertion, HEMS sorts the retrieval cue through the hierarchy, but does not merge or update the existing distributions in the memory models. HEMS implements loopy belief propagation over an approximation of the retrieved event memory element, known as a Bethe cluster graph [5], to infer the posterior network. Conditional samples are drawn from this by sampling the posterior network in a top-down manner.

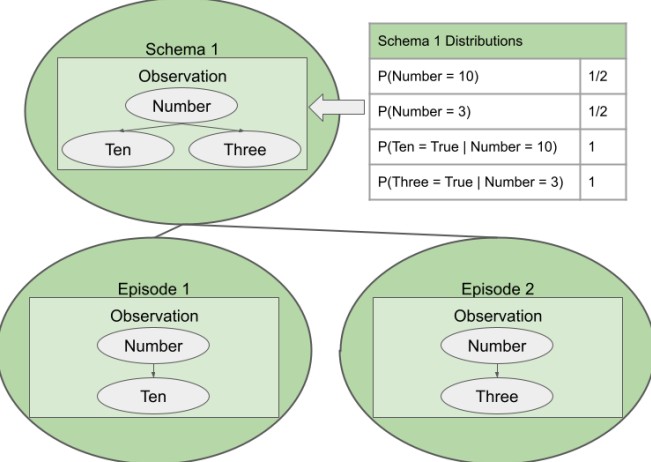

Figure 1: A notional generalization tree showing one schema node at the root and two children episodes. The distributions in the schema node probabilistically summarize its children.

## 3  Representing Temporal Episodes (1pg)

This work extends the HEMS event memory representation to include temporal episodes and schemas, which enables the system to model dynamics. HEMS temporal models are directed acyclic state transition graphs modeled as Bayesian networks. They include state, observation, and action nodes. Figure 2 shows a sample temporal schema. The schema elements define a state transition model $P(S_i|S_{i-1}, A_{i-1})$, such that $S_i$ is an arbitrary state and $A_i \in \mathbf{A}$, coming from the set of possible actions. The schema also includes an observation model, $P(O_i|S_i)$ and policy $\pi = P(A_i|O_i)$ for directing the agent's behavior. To accommodate the HEMS temporal models into the event hierarchy, we created three separate partitions to organize the event memory contents. As Figure 2 shows, the first partition, (green), stores and organizes observations, the second (red) stores states, and the third (blue) stores temporal models.

Importantly, state and observation nodes in temporal models are *content free*. They range over a set of pointers, $R$, that reference state and observation schema in the event hierarchy. This makes conditioning the temporal models simple because observations and actions are only conditioned on one variable, $R$, and yet abstracting the temporal model in this way permits HEMS to model arbitrarily complex dynamical systems.

To build the event hierarchy, we assume HEMS receives an observation, the state of the world, and the action taken in that moment. It inserts the observation and state episodes into memory, and returns pointers to the location where these models were inserted. HEMS uses these pointers to build the temporal episode such that the state and observation nodes point to the respective locations of the state and observation episodes. The action node simply captures the given action. Once the temporal episode is complete, as determined by a user-specified length, HEMS inserts it into memory as described in Section 2.

During retrieval and inference, we assume HEMS does not have access to the state. Given a (partial) observation, HEMS returns a pointer to the most similar element in memory. Following this, HEMS builds a *temporal retrieval cue* whose observation node contains the returned pointer. HEMS retrieves the best-matching temporal model and performs inference to produce a posterior temporal model conditioned on the contents of the retrieval cue. Then, HEMS follows the pointers in the posterior network to reveal the state and observation models. HEMS does a second inference step in each model to produce posterior estimates for state and observation variables. Conditional samples are drawn by querying this posterior network in a top-down manner.

## 4  Experimental Design and Empirical Evaluations

To evaluate our extensions to HEMS in sequential decision-making domains, we designed HEMS-BC, a behavior cloning agent with event memory capabilities. We conducted our experiments in two gymnasium [12] toy text domains, *FrozenLake* and *CliffWalking*, which are discrete action and observation space domains. In *FrozenLake*, the agent attempts to navigate a frozen lake to reach a goal position without falling through holes interspersed on the surface. The lake is slippery which causes the agent to randomly move one space in a perpendicular direction to the one commanded.

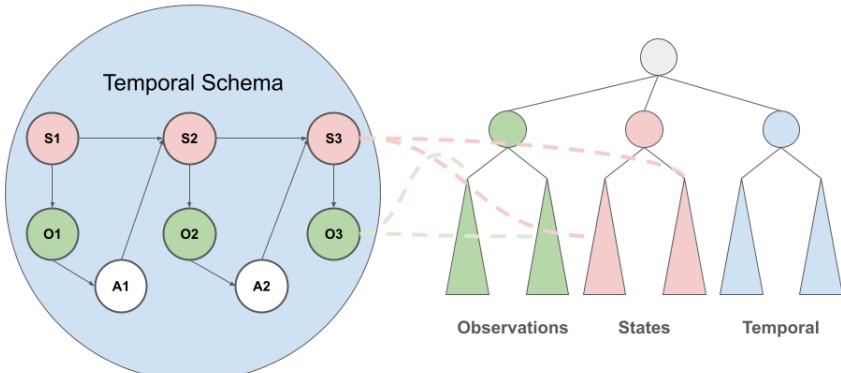

Figure 2: A sample temporal schema that includes state transition, observation, and policy.

The agent receives $+1$ reward for reaching the goal position and $+0$ otherwise. In *CliffWalking*, the agent moves around a cliff to reach the other side. The agent receives $-1$ reward for each time step the agent is not at the goal position and $-100$ reward if the agent steps over the cliff.

For both domains, HEMS represents the state by the agent's current row and column coordinates. Similar to Figure 1, state models include a random variable, $Number_i$ for $i \in [1, 2]$, for each coordinate and range over the possible values of that coordinate. Under this node exist binary-valued child nodes labeled with the written name of the number that takes value "$True$" when $Number_i$ takes its corresponding numeric value. Both domains represent the observation with a single variable, so parameterization is as shown in Figure 1.

For both domains, we repeated our experiments with five different random seeds. For each seed, HEMS-BC observed a variable number of expert demonstrations, adding each observation to the event memory. HEMS-BC generated a balanced training dataset by retrieving and sampling temporal models conditioned on the observations from the expert traces. We compared the performance of HEMS-BC in an ablation study with a BC baseline system with no event memory.

Figures 3a and 3d show that HEMS-BC approximates the expert policy from fewer expert demonstrations compared to the baseline BC agent. Figure 3a shows that HEMS-BC's performance monotonically improves and maintains expert-level performance, while the baseline system does not. Figure 3d shows that HEMS-BC reached expert performance after a single expert demonstration, while the baseline system needed 100 samples of the same trajectory. Lastly, Figures 3b and 3e show that HEMS-BC converges slightly faster during training, but each epoch takes roughly twice as long compared to the baseline performance, shown in Figures 3c and 3f.

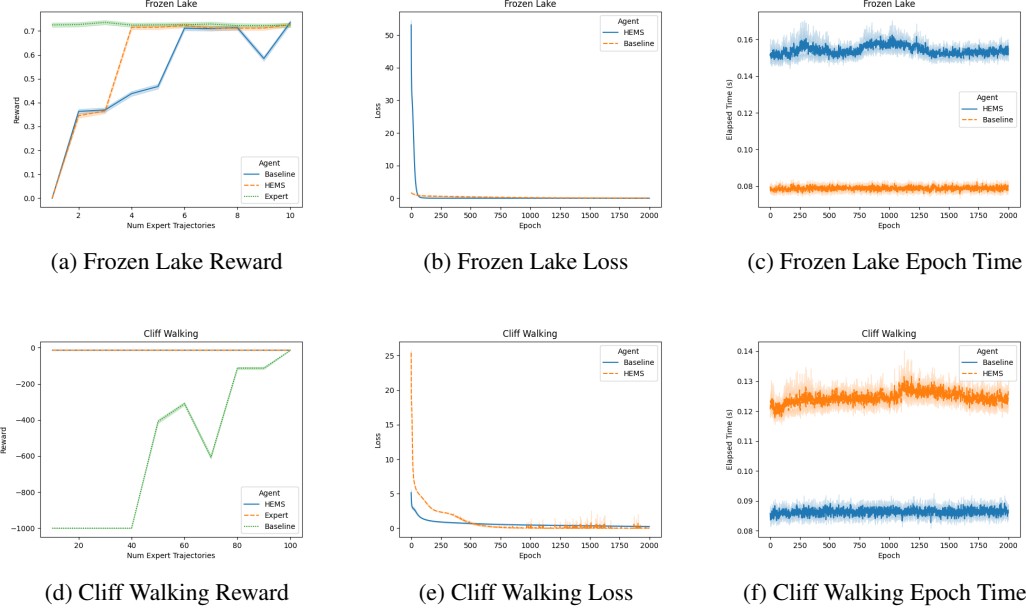

(a) Frozen Lake Reward      (b) Frozen Lake Loss      (c) Frozen Lake Epoch Time

(d) Cliff Walking Reward      (e) Cliff Walking Loss      (f) Cliff Walking Epoch Time

Figure 3: Our results show that HEMS-BC learns the expert policy in both domains from fewer expert demonstrations compared to the baseline BC system with no event memory capabilities.

## 5   Future Work and Conclusions

In this paper, we demonstrated how event memory capabilities improve sample efficiency for behavior cloning agents. Our agent, HEMS-BC, leveraged the hybrid event memory system to store and parametize the state, observation, and environmental dynamics as Bayesian networks. This permitted the agent to automatically rectify distributional biases in the data. We discussed how our novel temporal models enabled this by abstracting the dynamics from the state and observation models. Looking forward, we plan to extend HEMS-BC to learn hierarchical temporal models that represent higher-level procedural operations. In summary, this work provides a foundation for continued exploration in sequential decision-making in uncertain dynamical systems.

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
