# OpenReview forum: "Sample-Efficient Behavior Cloning using Hierarchical Event Memory of Bayesian Networks"
_NeurIPS.cc/2024/Workshop/BDU — Submitted to NeurIPS BDU Workshop 2024_

### Official Review · Reviewer_Xf5Q · 2024-10-03
**Sample-Efficient Behavior Cloning using Hierarchical Event Memory of Bayesian Networks**

**Rating:** 5
**Confidence:** 5

**Review:**

This paper proposes a novel approach called HEMS-BC for behavior cloning with improved sample efficiency.

Pros:

Psychologically-inspired memory model: HEMS-BC leverages the Hybrid Event Memory System (HEMS) to represent states, observations, and state transitions as Bayesian networks. This approach aligns with how humans remember and generalize procedures.
Addressing distributional bias: By retrieving and sampling temporal models conditioned on observations, HEMS-BC tackles the issue of class imbalance in behavior cloning datasets.
Improved sample efficiency: The paper demonstrates that HEMS-BC achieves expert-level performance with fewer expert demonstrations compared to a baseline system.
Clear experimental setup: The text environments (FrozenLake, CliffWalking) and evaluation metrics (reward, loss, training time) are clearly defined.

Cons:

Limited details on HEMS: While the paper mentions the theoretical underpinnings of HEMS, it lacks specifics on the insertion and retrieval algorithms within the hierarchical structure.
Ablation study limitations: The baseline system lacks event memory capabilities entirely, which might not be a strong comparison for evaluating the effectiveness of HEMS.
Computational cost: The paper acknowledges that HEMS-BC takes longer per epoch compared to the baseline, but a quantitative comparison of training times would be beneficial.

Originality:

The combination of Bayesian networks with a hierarchical event memory for behavior cloning is novel.
The concept of abstracting observation and state content from the environmental dynamics using content-free nodes is innovative.

Significance:

This work contributes to more data-efficient behavior cloning methods, which is crucial for real-world applications where obtaining expert demonstrations can be expensive.
The psychologically-inspired memory model offers a potentially more interpretable approach to behavior cloning compared to purely deep learning methods.

---

### Official Review · Reviewer_YiKp · 2024-10-06

**Rating:** 3
**Confidence:** 4

**Review:**

This paper introduces HEMS-BC that integrates Bayesian networks for memory representation, offering a more structured approach to modeling state transitions and actions.

Weaknesses:
The empirical evaluation is limited to discrete and small grid worlds, where there is limited need to use HEMS. This is perhaps due to the scalability weaknesses of the method. Also, there is a lack of comparison with other seminal BC methods such as GAIL. In addition, the novelty is very limited as the method simply uses HEMS with BC, and there is a lack of theoritical gounds.

---

### Decision · Program_Chairs · 2024-10-09

**Decision:**

Reject

**Comment:**

Review scores for this paper are mixed, tending to negative. Reviewers complain about narrowness, and worry that the method is in principle most useful in the same settings where it is not needed due to good alternatives. Reviewers also worry about lack of evaluation, in particular compared to other baselines. My overall impression is that, while this work contains nice ideas, particularly its memory aspects, it needs a bit more time in order to make the comparisons more comprehensive and get it to a state that it is ready to present. I encourage the authors to do so and resubmit to a future workshop.